# Comparative proteomics reveals mechanisms that underlie insecticide resistance in Culex pipiens pallens Coquillett

**Chongxing Zhang**[1]*, **Qiqi Shi**[2], **Tao Li**[3], **Peng Cheng**[1], **Xiuxia Guo**[1], **Xiao Song**[1], **Maoqing Gong**[1]*

**1** Shandong Institute of Parasitic Diseases, Shandong First Medical University & Shandong Academy of Medical Sciences, Jining, Shandong, P.R. China, **2** National Institute of Parasitic Diseases, Chinese Center for Disease Control and Prevention, Key Laboratory of Parasite and Vector Biology, MOH, National Center for International Research on Tropical Diseases, WHO Collaborating Centre for Tropical Diseases, Shanghai, China, **3** Nanning MHelixProTech Co., Ltd., Nanning Hi-tech Zone Bioengineering Center, Nanning, P.R. China

* chongxingzhang@aliyun.com (ZCX); gmq2005@163.com (GMQ)

**Data Availability Statement:** Data can be found in Dryad Digital Repository https://datadryad.org/stash/dataset/doi:10.5061/dryad.pzgmsbcjs.

## Abstract

Mosquito control based on chemical insecticides is considered as an important element of the current global strategies for the control of mosquito-borne diseases. Unfortunately, the development of insecticide resistance of important vector mosquito species jeopardizes the effectiveness of insecticide-based mosquito control. In contrast to target site resistance, other mechanisms are far from being fully understood. Global protein profiles among cypermethrin-resistant, propoxur-resistant, dimethyl-dichloro-vinyl-phosphate-resistant and susceptible strain of *Culex pipiens pallens* were obtained and proteomic differences were evaluated by using isobaric tags for relative and absolute quantification labeling coupled with liquid chromatography/tandem mass spectrometric analysis. A susceptible strain of *Culex pipiens pallens* showed elevated resistance levels after 25 generations of insecticide selection, through iTRAQ data analysis detected 2,502 proteins, of which 1,513 were differentially expressed in insecticide-selected strains compared to the susceptible strain. Finally, midgut differential protein expression profiles were analyzed, and 62 proteins were selected for verification of differential expression using iTRAQ and parallel reaction monitoring strategy, respectively. iTRAQ profiles of adaptation selection to three insecticide strains combined with midgut profiles revealed that multiple insecticide resistance mechanisms operate simultaneously in resistant insects of *Culex pipiens pallens*. Significant molecular resources were developed for *Culex pipiens pallens*, potential candidates were involved in metabolic resistance and reducing penetration or sequestering insecticide. Future research that is targeted towards RNA interference of the identified metabolic targets, such as cuticular proteins, cytochrome P450s, glutathione S-transferases and ribosomal proteins proteins and biological pathways (drug metabolism—cytochrome P450, metabolism of xenobiotics by cytochrome P450, oxidative phosphorylation, ribosome) could lay the foundation for a better understanding of the genetic basis of insecticide resistance in *Culex pipiens pallens*.

**Funding:** GMQ and ZCX was funded by the National Natural Science Foundation of China (NSFC) (Grant No. 8187168), GMQ, ZCX and CP were supported through Academic promotion programme of Shandong First Medical University (2019QL005), ZCX was supported by the development of medical science and technology project of Shandong Province (2018WS302). The funders had no role in study design, data collection and analysis, decision to publish, or preparation of the manuscript.

**Competing interests:** I have read the journal's policy and the authors of this manuscript have the following competing interests: Author Tao Li was employed by the company Nanning MHelix ProTech Co., Ltd., the remaining authors declare that the research was conducted in the absence of any commercial or financial relationships that could be construed as a potential conflict of interest.

## Author summary

Global protein profiles were compared among a susceptible strain of *Cx. pipiens pallens* and strains that were cypermethrin-resistant, propoxur-resistant, and dimethyl-dichloro-vinyl-phosphate-resistant after 25 generations of selection by distinct chemical insecticide families, multiple mechanisms were found to operate simultaneously in resistant mosquitoes of *Cx. pipiens pallens*, including mechanisms to lower penetration of or sequester the insecticide or to increase biodegradation of the insecticide via subtle alterations in either the cuticular protein levels or the activities of detoxification enzymes (P450s and glutathione S-transferases).

## Introduction

Mosquitoes and mosquito-borne diseases continue to pose a huge public health burden worldwide; malaria, lymphatic filariasis, dengue, chikungunya, and West Nile virus cause significant medical and economic impacts that disproportionately affect developing countries[1–3].

Since the 1950s, chemical insecticides have been used on a massive scale to control mosquito populations. However, long-term intensive and widespread overuse or misuse use of insecticides has applied intense selection pressure that has led to the development and subsequent intensification of various genetically modulated resistance mechanisms in mosquitoes [4–6] and has resulted in a rise in mosquito-borne diseases and outbreaks of mosquito-related diseases in recent years[7,8]. Today, with the widespread development of resistance in mosquitoes to the most commonly used insecticides, insecticide resistance is regarded as the most serious threat to the control of mosquitoes and mosquito-borne diseases[9]. Vector control of mosquitoes remains crucial to reduce disease transmission, and has long been a critical part of the global strategy to manage mosquito-associated diseases; the application of chemical insecticides is the most important component in mosquitoes control[10], and thus characterizing the molecular mechanisms underlying resistance is a key step for improving resistance management strategies.

Previous studies on insecticide resistance makes it possible to classify insect adaptations, abiotic factors, such as mosquito larvae feeding on plant debris that accumulated insecticide residues, or grow in water bodies enriched with plant compounds generate tolerance to insecticides or promote detoxification pathways of these insecticides against mosquitoes[11], biotic vary from symbionts (microbes living in the mosquitoes) to entomopathogen opportunistic organisms[12] are able to affect detoxification systems[13] or leading to the appearance of insecticide resistance[12]. The molecular mechanisms of insecticide resistance have been performed at the transcriptional level through cDNA microarray[14,15], RNA-seq[16–19] and proteomic[20–25] analyses, leading to exciting progress with regard to the transcriptional basis of insecticide resistance, the physiological and biochemical mechanisms of insects develop resistance to insecticides through four trajectories: target-site resistance (mutation of the protein targeted by the insecticide, that is knockdown resistance, kdr), metabolic resistance (increase rate of detoxification of insecticides), penetration resistance (thicker cuticle for decreased entry of insecticides), and sequestration of the insecticide[26,27]. A common phenomenon of insecticide resistance is that multiple mechanisms operate simultaneously in resistant insects such as the house fly[28–30], cockroach[31], mosquito[32–35], cotton bollworm[36], and bed bug[37].

To date, there are two main types of molecular mechanisms of mosquito vectors can develop resistance to insecticides included target-site insensitivity and metabolic resistance [27,38]. Dramatic progress has been made in identifying target-site insensitivity in mosquitoes, including *Anopheles gambiae* [39–41], *An. arabiensis*[42], *Aedes albopictus* [43], and *Cx. pipiens* [41,44]. The metabolic detoxification is an acquired resistance mechanism, which is regulated by the activity of certain oxidized enzymes, has been reported worldwide, cytochrome P450 monooxygenases (P450s or CYPs), glutathione S-transferases (GSTs) and carboxy/choline esterases (CCEs) are known for their roles in insecticide metabolism in insects [45,46]. Cytochrome P450s[38,47,48], GSTs[49], and UDP glucosyl-transferases (UGTs)[50] overproduction have been frequently associated with resistance to chemical insecticides in mosquitoes[26,27]. Other less common mechanisms that develop resistance in insects are the resistance per behavior and the resistance per decreased penetration through the cuticle or cross resistance[51]. Typically, a combination of diverse mechanisms provides significantly higher levels of resistance than one individual mechanism[52]. Due to the large number of mosquito genes encoding detoxification enzymes[53–55], pinpointing those responsible for resistance remains challenging[26].

Proteins, as the primary functional molecules, are the executors that function in many different physiological processes, proteomic analysis is an effective, powerful method for gaining insight into the mechanisms of insecticide resistance at the molecular level[20–25], currently, large-scale expression profiling and screening for resistance-related proteins in mosquitoes are limited and include two-dimensional gel electrophoresis (2-DE) proteomics[56,57], but few attempts have been made to apply proteomic analysis technique to study the insecticide resistance of mosquitoes iTRAQ[20]. Today, with the advent of highly advanced proteomic platforms based on isobaric tags for relative and absolute quantification[58], such knowledge gaps can be overcome by high-throughput sequencing approaches thatcan generate concomitant protein expression and ultimately provide direct insight into the activity of relevant proteins.

In the present study, iTRAQ labeling coupled with liquid chromatography/tandem mass spectrometric (LC-MS/MS) analysis was used to determine changes in protein levels associated with adaptation to three insecticides (the pyrethroid cypermethrin, propoxur, and dimethyl-dichloro-vinyl-phosphate) from distinct chemical families in the mosquito *Cx. pipiens pallens*. The results are discussed with regard to known and new putative adaptive mechanisms conferring insecticide resistance in mosquitoes, which improves our understanding of the mechanisms developed by mosquitoes to resist insecticides.

## Methods

### Ethics statement

All animal experiments were carried out in accordance with the Animal Protection Law of the People's Republic of China and with the approval of the Ethical Committee of Shandong First Medical University & Shandong Academy of Medical Sciences (Jinan, Shandong).

### Mosquito selection with insecticide and larvicidal bioassay

The laboratory Tangkou strain of *Cx. pipiens pallens*, originating from Tangkou Village (Jining prefecture, Shandong Province), has been kept in colonies in standard insectary conditions without exposure to any insecticides since 1960 and thus is fully susceptible to insecticides. All the susceptible mosquito populations were reared at a constant room temperature of approximately 28°C and 75% relative humidity with a photophase of 14 h and a scotophase of 10 h. Adult mosquitoes were provided with 10% sugar solution and were used as the parental strain to repeatedly selected for 25 generations to produce three independent resistant strains at the

larval stage with resistance to the pyrethroid insecticide cypermethrin, the carbamate insecticide propoxur and the organophosphates insecticide DDV at the median lethal concentration (LC$_{50}$)[20]. The detailed selection procedure was described previously[59,60]. Technical-grade formulations of cypermethrin, propoxur and DDV were employed for the larval bioassay study. Different concentrations of these insecticides were prepared using denatured alcohol as solvent (98 ml of absolute alcohol + 2 ml of methyl ethyl ketone) as described by Shetty[61]. The cypermethrin-, propoxur- and DDV-selected strains were defined as cypermethrin-resistant (Cx_cym), propoxur-resistant (Cx_pro) and DDV-resistant (Cx_ddv) strains, respectively.

For validation of different protein expression in the midgut of *Cx. pipiens pallens*, larvae were reared in round plastic tubs (diameter 0.6 m) filled with water with fish food twice daily. Experimental larvae were randomly collected from several tubs to compensate for size differences and feeding history which are known to be influenced by larval density. All laboratory experiments were carried out with third instar larvae laboratory-reared larvae. Fifty third instar larvae were exposed for 5 h to 0.021 mg/l of *Bacillus thuringiensis* subspecies *israelensis* (bti, mosquitoes larvae that exposed to bti refer as Cx_bti, Shandong Lukang Sheryl Pharmaceutical Co., Ltd, 3000 international toxic units [ITU/mg]), with the dosage following the standard testing procedures for microbial tests [62]. We chose a 5 h treatment time to allow larvae to ingest the bti without severely affecting their behavior and without causing evident damage to the intestinal tissue at the microscopic histological level. In the control group, larvae were exposed to water under the same conditions (refer as Cx_nbti). Larvae exposed to the insecticide were washed in water and midgut pools were dissected and transferred to 1.5-ml microcentrifuge tubes containing 1mm phenylmethylsulfonyl fluoride (PMSF) and stored at −80˚C before use.

## Protein extraction

Each sample (the Cx_cym, Cx_pro, Cx_ddv and Cx_s strains) was extracted separately from 5 egg rafts, 5 fourth-instar larvae, 5 pupae, five 2- to 3-day-old adult males, 5 adult females (without blood feeding) and midgut pools of *Cx. pipiens pallens*. Briefly, each samples was completely homogenized in protein extraction buffer (8 M urea, 2 mM EDTA, 10 mM dithiothreitol (DTT) and 1% Amresco Protease Inhibitor Cocktail). Then, samples were lysed at room temperature with 3 min of sonication and centrifuged for 10 min at 13000 rpm, the supernatants were precipitated with chilled acetone for 3 h at -20˚C.

## Quantification of proteins

The resulting protein concentration of each sample was measured using a Bradford Protein Assay Kit. Then, a total of 100 μg protein/sample was combined from the egg, larvae, pupae, and adult male and female stages (20 μg each) and reduced with 10 mM DTT for 60 min at 37˚C. This was followed by the addtion of 25 mM iodoacetamide (IAA) to each protein samples and then alkylation for 45 min in darkness at room temperature. Subsequently, 100 mM tetraethylammonium bromide (TEAB) was added to reduce the concentration of urea to $< 2$ M.

## Trypsin digestion

Finally, samples were digested with trypsin (enzyme-to-substrate ration of 1:50) overnight at 37˚C, and an additional second digestion was done for 4 h to ensure complete cleavage.

## Samples desalting

After trypsin digestion, each peptides samples was desalted using a Strata X SPE column, dried, and resuspended in 25 μL 500 mM TEAB, and labeled with an 8-plex iTRAQ kit. Each dried and labeled peptide sample was reconstituted using High Performance Liquid Chromatography (HPLC) solution A (2% acetonitrile [can], pH 10) and fractionated by high-pH reverse-phase HPLC on a Waters Bridge Peptide BEH C18 (130 Å, 3.5 μm, 4.6 × 250 mm). Loaded peptides were eluted with 2% to 98% acetonitrile gradient buffer solution at pH 10 in 60 fractions at a speed of 0.5 ml/min over 88 min. A total of 20 fractions were combined and each fraction was desalted by using ZipTip C18 tips (Merck Millipore, Ziptip Pipette Tips 10μL). Sample fractions were dried on a vacuum concentrator and stored at -20˚C pending MS analyses.

## High-resolution LC-MS/MS analysis

LC-MS/MS experiments were performed on a NanoLC 1000 LC-MS/MS liquid chromatography system (Thermo) equipped with a Proxeon EASY-nLC 1000 coupled to an LTQ-Orbitrap Elite. Briefly, peptides were acidified in 0.1% formic acid and trapped in a 3 μm 100 Å Acclaim 100C18 capillary liquid chromatographic column (Thermo) 75 μm×2 cm in 100% solvent A (0.1 M acetic acid in water) at a flow rate of 5 μL/min. Then, analytical separation was performed on 2 μm 100 Å Acclaim C18 capillary liquid chromatographic column (Thermo) 50μm × 15 cm using a linear gradient of solvent B (98% ACN with 0.1% formic acid). The gradient ran at a flow rate of 250 nl/min as follows: 0–60 min from 10% to 35% solvent B, 35% to 50% in 10 min and then ramping to 100% solvent B. Mass spectrometry was operated in a data-dependent mode, automatically switching between MS and MS/MS (at mass range of m/z 350 to 1800 and resolution of 60,000) using repetitive full MS spectra scans followed by collision-induced dissociation (CID, at 38 normalized collision energy). The 20 most intense precursors were selected for subsequent decision tree-based ion trap CID fragmentation in the MS survey scan, and the following settings were used: collision energy 38% above threshold, ion count 300 with 30.0 s dynamic exclusion, full width at half maximum (FHMW) 400 m/z using an AGC setting of 1e6 ions, and fixed first mass was set as 100 m/z.

## Mass spectrometry data analysis

The mass spectrometry MS/MS raw data were searched against all *Culex quinquefasciatus* protein in the *Culex* Taxonomy database (S1 Data), downloaded from the UniProt Database using SEQUEST software integrated in Proteome Discoverer (version 1.3, Thermo Scientific) with the following parameters: mass tolerance set to 20 ppm, product ion tolerance set to 0.02 Da, and only trypsin sequences allowed. Oxidation and protein N-terminal acetylation were accepted as variable modifications and carbamidomethylation was accepted as a static modification. A maximum of two miscleavages was permitted, and peptide- and protein-level false discovery rates (FDRs) were filtered to 1%.

## Functional classification of proteins

Functional annotation and classification of all identified proteins were determined using the Blast2GO program against the nonredundant NCBI protein database. The gene function databases compiled for the Gene Ontology (GO), the Kyoto Encyclopedia of Genes and Genomes (KEGG) pathways, and Cluster of Orthologous Groups (COG) were used to infer the putative functions of the identified proteins. Detailed information can be found on at http://www.geneontology.org. Significant GO terms ($p < 0.05$) were mapped with the REVIGO online

tool (http://revigo.irb.hr), which removes redundant GO terms and visualizes the semantic similarity of remaining terms[63]. KEGG pathway analysis is part of the Kyoto Encyclopedia of Genes and Genomes database, which is a reference database for pathway mapping. For KEGG pathway analyses, the main biochemical metabolism and signal transduction pathways of the identified proteins were extracted using the Search pathway tool on the KEGG Mapper platform (http://www.genome.jp/kegg/mapper.html) [64–66]. COG is the database for orthologous protein classification. Identified proteins were compared with the COG protein database to predict the functions of proteins and conduct statistical analyses.

### Functional enrichment, protein expression pattern and function clustering

GO, KEGG pathway and domain enrichment statistics were calculated using Fisher's exact test: correction for multiple hypothesis testing was carried out with standard false discovery rate control methods and the pathways with a corrected $p$ value $< 0.05$ were considered significant pathways.

Expression-based and functional enrichment-based clustering for different protein groups was used to explore the potential relationships between different protein groups with special protein functions. A heatmap was generated by the software Heml 1.0.3[67]. The expression pattern of the protein was completed by using the timeclust function in the R package 'TCseq'. First, we collated all of the protein groups obtained after functional enrichment analysis along with their $p$ values. Second, we sorted those categories enriched in at least one of the protein groups with a $p$ value $<0.05$. This filtered $p$ value matrix was transformed by the function x = −log10 ($p$ value). Third, z-transformation was applied to x values for each functional category, and Z-scores were clustered by one-way hierarchical clustering (Euclidean distance, average linkage clustering). Finally, cluster membership was visualized by a heat map using the "pheatmap" function from the R package. Prediction of protein-protein interactions (PPIs) was done using the STRING database (version 10.0) and visualized using Cytoscape software (version 3.4.0)[68,69]. We selected the combined score $> 0.400$ between nodes for PPI with Cytoscape (version 3.0), part of which is shown in the paper but only containing thefor important proteins.

### Parallel reaction monitoring analysis

A label-free, targeted parallel reaction monitoring (PRM) method[70] was adopted to verify the reliability of our label-based proteomics. A total of 62 differentially expressed proteins, including one internal standard housekeeping protein, were chosen. For the Cx_cym, Cx_pro, Cx_ddv and Cx_s strains, equal amounts of protein samples were analyzed for semiquantitative measurements, and each strain underwent three replications. Peak areas were extracted from PRM mass spectrum data using Skyline software[71].

## Results

### Insecticide resistance levels

After 25 generations of larval selection with the insecticides cypermethrin, propoxur and DDV, bioassays revealed a constitutively increased resistance of each selected strain to its respective insecticide compared to the susceptible parental strain (S1 Table). The resistance levels of the Cx_pro and Cx_ddv strains were moderate but significant (11.34-fold and 8.17-fold respectively). Although significant, the resistance level of the Cx_cym strain to cypermethrin susceptible was considerably higher (243.00-fold).

## Protein identification and differentially expressed proteins (DEP) screening

A total of 2,502 protein were identified in two replicates of the LC-MS/MS experiments with a high confidence of peptide selection (FDR = 0.01). Among them, 1,513 proteins achieved quantitative significances, all the proteins were completely annotated with the accession number for each identified protein arranged by the Gene Ontology, KEGG pathway, Domain, and Cluster of Orthologous Groups (COG) function catalogues and Subcellular Location analyses (S2 Table). Our analysis revealed significant changes in proteins in these Cx_cym, Cx_pro, Cx_ddv, and Cx_s strains (S3 Table), there were 164 proteins with significant differential expression (fold change > 1.2 and $p$ value < 0.05) between the Cx_cym and Cx_s control groups, including 54 upregulated and 110 downregulated proteins. A total of 156 proteins changed dynamically between the Cx_pro and Cx_s control groups, including 59 upregulated and 97 downregulated proteins, whereas 81 proteins changed between the Cx_ddv and Cx_s control groups, with 31 up- and 50 downregulated proteins. The repeatability analysis showed a considerably high correlation between both biological replicates, demonstrating the reliability of our experimental protocol (Fig 1).

## Bioinformatics analyses of the altered proteins

To better understand the characteristics of the proteins altered in response to insecticide selection, GO ontology analysis was performed to identify the upregulated and downregulated proteins among the Cx_cym, Cx_pro, Cx_ddv, and Cx_s strains of *Cx. pipiens pallens* (Figs 2 and 3). As shown, the most enriched GO terms regarding the molecular functions of the upregulated proteins were structural constituent of cuticle, structural molecule activity, lipid

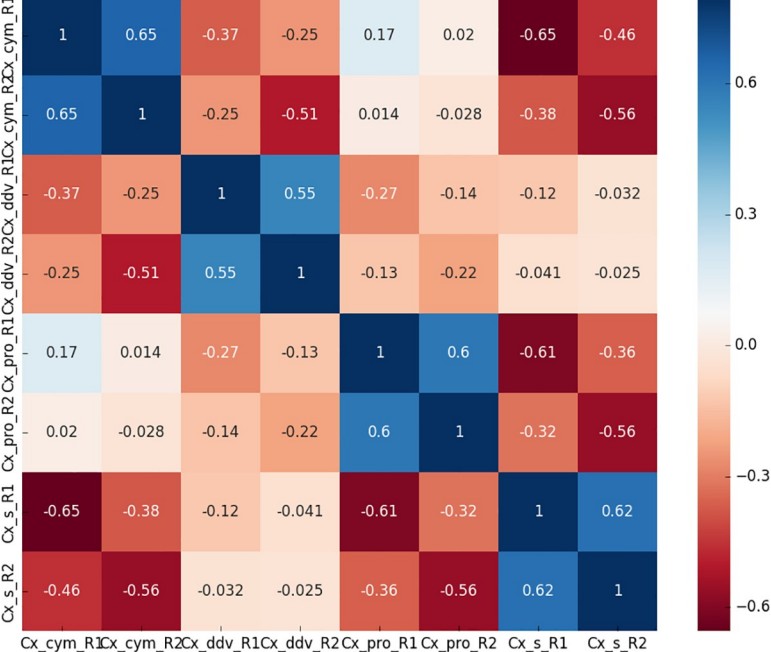

**Fig 1. Repeatability analysis between two replicates of LC-MS/MS experiments of *Cx. pipiens pallens* in response to three insecticides.** The figure shows correlation between two replicates of LC-MS/MS samples, number in the square frame means that repeatability of differentially expressed proteins between the samples, the closer correlation value to 1 between the LC-MS/MS samples, the same samples tend to be, dark blue in the figure.The smaller the value, the more statistical significance it is.

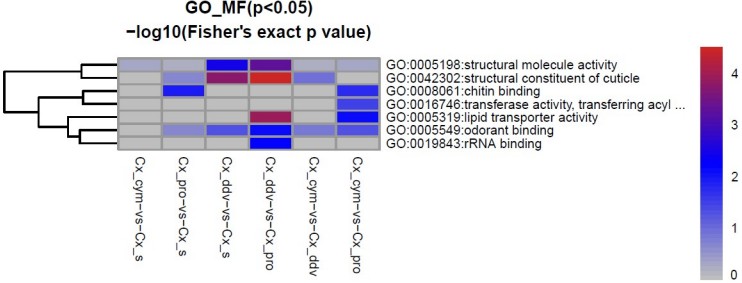

**Fig 2. Significantly upregulated GO molecular function cluster in *Cx. pipiens pallens* in response to three insecticides.** Upregulated proteins in the molecular function category according to the Fisher's extract test, chitin binding was upregulated by comparing Cx_pro vs.Cx_s; structural constituent of cuticle and structural molecule activity were upregulated by comparing Cx_ddv vs.Cx_s; structural constituent of cuticle, lipid transporter activity and structural molecule activity were upregulated by comparing Cx_ddv vs. Cx_pro; lipid transporter activity and chitin binding were upregulated by comparing Cx_cym vs.Cx_pro. Scale bar is the value calculated by—log 10 (Fisher exact p value), distinguished by color. The larger the value, the more red it is. The smaller the value, the more blue it is, Fisher exact p value is used to measure statistical significance.

transporter activity, chitin binding, transferase activity, transferring acyl groups and odorant binding, which had higher fold enrichment (log2) and Fisher's exact test p-values (-log10). Among the downregulated proteins, structural constituent of cuticle was the most enriched GO term.

In KEGG enrichment analysis (S1 Fig), metabolism of xenobiotics by cytochrome P450 (map00980) and drug metabolism—cytochrome P450 (map00982) pathways respond to insecticide selection conditions, implicating P450s as the important regulatory enzymes associated with metabolic resistance in *Cx. pipiens pallens* under insecticide selection. Furthermore, the differential changes in glutathione S-transferase (B0W6D0), microsomal glutathione s-transferase

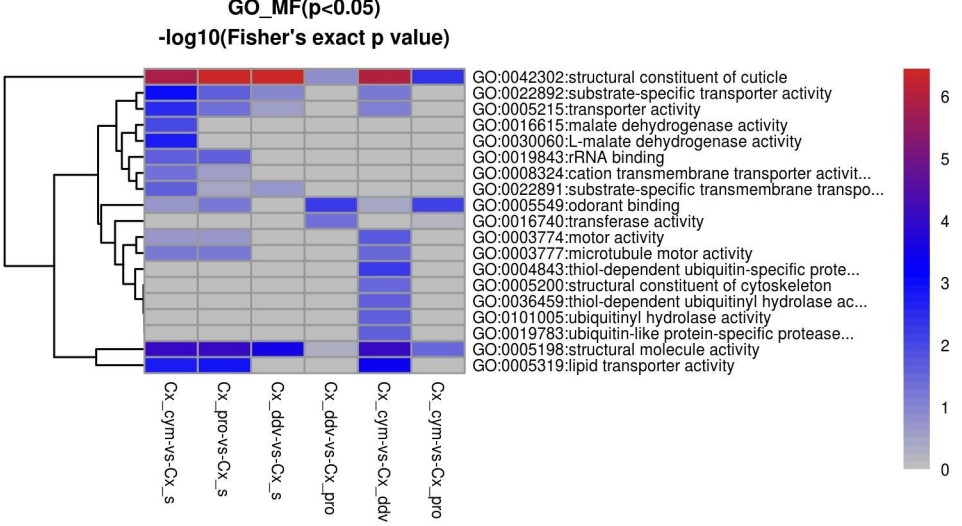

**Fig 3. Significantly downregulated GO molecular function cluster in *Cx. pipiens pallens* after three insecticides selection.** Downregulated proteins in the molecular function category according to the Fisher's extract test, structural constituent of cuticle and structural molecule activity were downregulated by comparing Cx_cym, Cx_pro, Cx_ddv vs. Cx_s and Cx_cym vs. Cx_ddv, respectively; odorant binding and transferase activity were downregulated by comparing Cx_ddv vs. Cx_pro; structural constituent of cuticle was downregulated by comparing Cx_cym vs. Cx_pro. Scale bar is the value calculated by—log 10 (Fisher exact p value), distinguished by color. The larger the value, the more red it is. The smaller the value, the more blue it is, Fisher exact p value is used to measure statistical significance.

(B0X075) and glutathione-s-transferase theta (B0XGK3) proteins in both pathways suggest a critical regulatory role associated with P450. These findings indicate that glutathione S-transferase may play an important role in the regulation of metabolic resistance pathway in *Cx. pipiens pallens*. The KEGG pathways annotation of comparison shows that oxidative phosphorylation (map00190) is activated associated with electron-transport chain subunits (Ndufa2, Ndufa4, Ndufa5, Ndufa7, Ndufa8, Ndufa9, Ndufab1, COX5), and ribosome (map03010) is activated, associated with S15e, L34e, S6e, and L18Ae. Interestingly, the quantities of these subunits deviated simultaneously towards different directions; Ndufs4 and Ndufs decreased while Ndufa4 increased. Oxidation phosphorylation and membrane-localized proteins are critical nodes to prevent malfunction in glycolytic flux and intracellular signal transit (S1 Fig).

Comparing the patterns of expression of different proteins among Cx_cym, Cx_pro, Cx_ddv, and Cx_s strains of *Cx. pipiens pallens*, all of these proteins are significantly differentially expressed in resistant strains when compared to their expression in the susceptible strain. Almost all protein enrichment changed at the peak, and the lowest valleys were in Cx_cym, Cx_pro and Cx_ddv, indicating that most of these proteins showed an increase or decrease in expression following the insecticide selection from susceptible to resistance strains in *Cx. pipiens pallens*. The insecticide selection alters the normal developmental pathway into an alternative one (S2 Fig).

In Cx_cym vs. Cx_s, the upregulated proteins were TGS-like, OBG-type guanine nucleotide-binding (G) domain, GTP binding domain, MD-2-related lipid-recognition domain, beta-grasp domain, and chaperonin Cpn60/TCP-1 family, whereas the downregulated proteins were chitin-binding type R&R consensus, insect cuticle protein, actin, conserved site, actin/actin-like conserved site, actin family, vitellogenin, and open beta-sheet. In Cx_pro vs. Cx_s, upregulated proteins were FAD dependent oxidoreductase, chitin binding domain, serine proteases, trypsin family, serine active site, serine proteases, trypsin family, histidine active site, peptidase S1A, and chymotrypsin family, whereas the downregulated proteins were insect cuticle protein, chitin-binding type R&R consensus, actin, conserved site, actin/actin-like conserved site, actin family, vitellogenin, and open beta-sheet. In Cx_ddv vs. Cx_s, upregulated proteins were immunoglobulin E-set, ribosomal protein L30, ferredoxin-like fold domain, ribosomal protein L30, conserved site, chitin-binding type R&R consensus, and hemocyanin, N-terminal, whereas the downregulated proteins were insect cuticle protein, chitin-binding type R&R consensus, adult cuticle protein 1, and insect odorant-binding protein A10/ejaculatory bulb-specific protein (S3 Fig).

### Prediction of the PPI network under insecticide selection

Because protein–protein interactions play important roles in most cellular biological processes, the PPI networks under insecticides selection conditions were predicted using the STRING (www.string-db.org) online PPI prediction software to further analyze the functional correlation of differentially expressed proteins between each group (S4–S6 Figs).

### Validation of proteomics data by parallel reaction monitoring

To validate the accuracy and reproducibility of the proteomics analysis results, PRM-quantifiable strategy analyses were performed, Sixty-two proteins were selected as those that may play important roles in insecticides resistance, and the differentially expressed levels of these proteins were quantified in Cx_cym, Cx_pro, Cx_ddv and Cx_s strains of *Cx. pipiens pallens*, Energy production and conversion, Putative uncharacterized protein (B0WRN8, B0W0Q9), NADH dehydrogenase 1 alpha subcomplex subunit 5 (B0WLS3), ubiquinol-cytochrome c reductase complex core protein (B0WXM0), pyruvate dehydrogenase (B0XA87), mitochondrial malate

dehydrogenase 2 (B0WMC8), malate dehydrogenase (B0W5T5), ATP synthase delta chain, mitochondrial (B0WYE7), NADH:ubiquinone dehydrogenase (B0WA75, B0W7N4), NADH-ubiquinone oxidoreductase B8 (B0WUY1), cytochrome c oxidase, -subunit VIb (B0WY92), NADH dehydrogenase iron-sulfur protein 8, mitochondrial (B0XGQ7). Translation, ribosomal structure and biogenesis, acidic ribosomal protein P1 (B0WU22), ribosomal protein L28 (B0WN15), 40S ribosomal protein S21 (B0W289), 60S ribosomal protein L10 (B0XGR8), ribosomal protein L37 (B0WQU6), putative uncharacterized protein (B0WIR5), 40S ribosomal protein S15 (B0W2E6). Lipid transport and metabolism, 3-hydroxyisobutyryl-CoA hydrolase, mitochondrial (B0WY25), acetyl-coa synthetase (B0X1B3), acetyl-CoA acetyltransferase, mitochondrial (B0XJ38), acetyl-coa carboxylase (B0WE67). Posttranslational modification, protein turnover, chaperones, thioredoxin reductase 1 (B0WGF9), proteasome endopeptidase complex (B0WTL4), T-complex protein 1 subunit delta (B0W5S5), ribosomal protein S6 (B0XH45), glutathione S-transferase 1–1 (B0XGJ8), glutathione S-transferase (B0VZ90), glutathione-s-transferase theta, gst (B0XGK0). Inorganic ion transport and metabolism, sodium/potassium-dependent ATPase beta-2 subunit (B0WIC2), sodium/potassium-dependent ATPase beta-2 subunit (B0WIC5), sodium/potassium-transporting ATPase subunit alpha (B0WFD0).

Cytoskeleton, actin (B0WY76), tropomyosin invertebrate (B0X3L6), myosin-Id (B0W191). Intracellular trafficking, secretion, and vesicular transport, secretory carrier-associated membrane protein (B0WUR5), bridging integrator (B0WUH3). Carbohydrate transport and metabolism, glyceraldehyde-3-phosphate dehydrogenase (B0WEB5), pyruvate kinase (B0WL21), triosephosphate isomerase (B0W5W4). Cell motility, thymosin 1 (B0X0W6). Amino acid transport and metabolism, clip-domain serine protease (B0W0Z4), pyrroline-5-carboxylate reductase (B0WZS2), hydrolase (B0X5I3), D-amino acid oxidase (B0WJC4). General function prediction only, leucine aminopeptidase (B0W6N8), charged multivesicular body protein 6-A (B0X3V8), Zinc finger CDGSH domain-containing protein 1 (B0WWX0), rhythmically expressed gene 2 protein (B0WVG3). Signal transduction mechanisms, troponin C (B0W6W0), phosphatase 2C beta (B0WCT7), GTP-binding protein 128up (B0WEM1). Nucleotide transport and metabolism, uridine phosphorylase (B0X890). Coenzyme transport and metabolism, S-adenosylmethionine synthase (B0W244).Secondary metabolites biosynthesis, transport and catabolism, cytochrome P450 4g15 (B0WQV0), cytochrome P450 (B0XDA9), and pupal cuticle protein (B0X2L1). Verification and validation of all proteins, peptides, and measured peptide peak areas are listed (S4 Table).

## Differential protein expression in the midgut of *Cx. pipiens pallens* induced by bti

A total of 564 proteins were identified, with 559 proteins achieving quantitative significance. There were 66 proteins with significant differential expression (fold change > 1.2 and *p* value < 0.05) between the Cx_bti and Cx_nbti control groups, including 33 upregulated and 33 downregulated proteins (S5 Table). Most of the proteins were cuticular and cytoskeleton related proteins, and metabolic enzymes are enriched in differential changes in the digestive tract linings dissected from Cx_bti strain of *Cx. pipiens pallens* (Figs 4 and 5 and S7).Given our interest in the role of P450s in metabolic resistance, we further profiled the differential protein expression of Cytochrome P450 3A19 (B0X758) in the midgut of third instar larvae of *Cx. pipiens pallens*, P450 protein showed higher expression levels in Cx_bti strains.

## Discussion

*Cx. pipiens pallens*, the primary vector of the Japanese encephalitis virus and filariasis in China, has developed resistance to several insecticides used for mosquito control. To date, the lack of

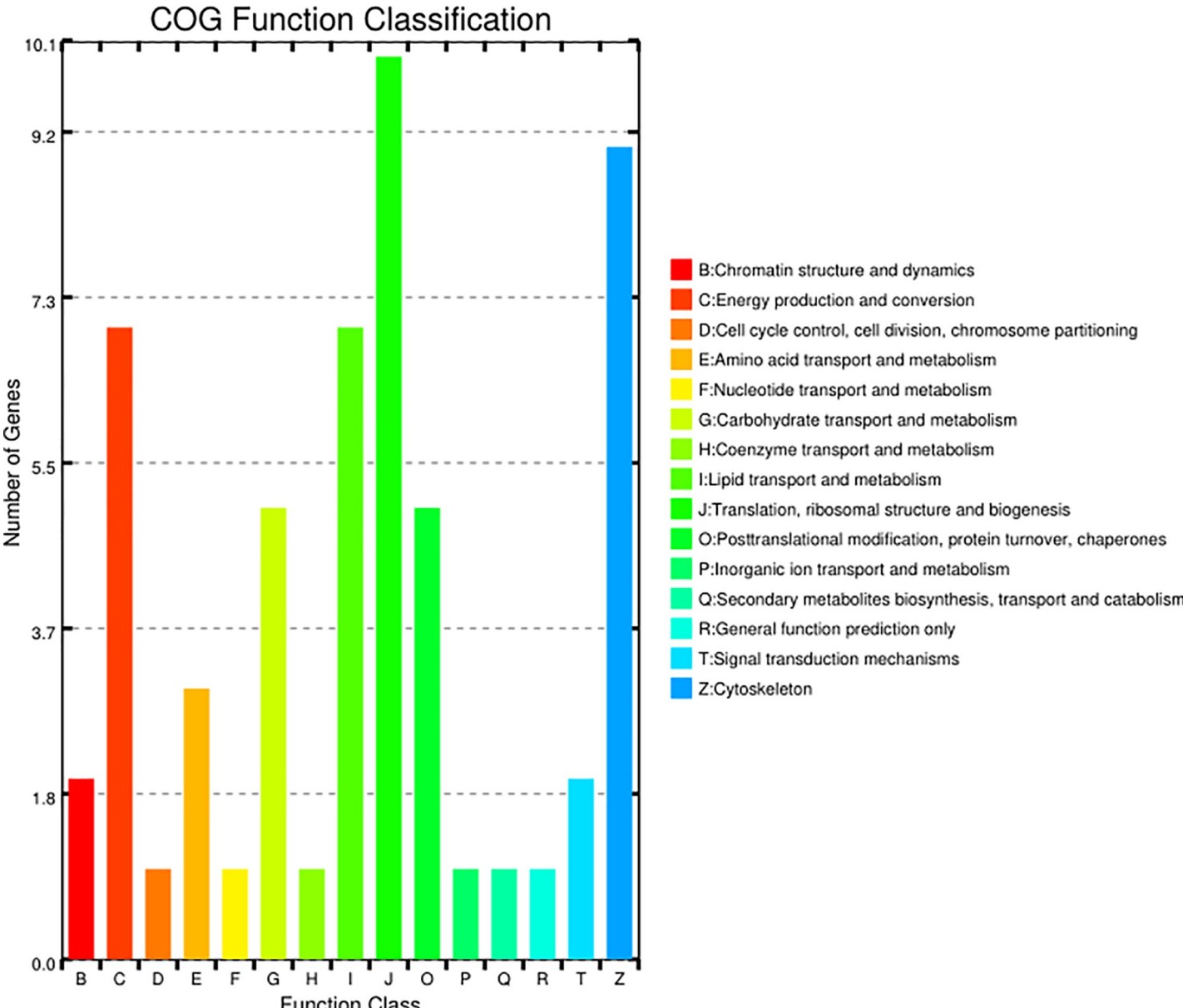

**Fig 4. Cluster of orthologous groups (COG) classification of putative proteins in the midgut of *Cx. pipiens pallens* exposed with bti.** 559 putativeproteins were classified functionally into 15 molecular familiesin theCOG database, the "J" and"Z" clusterrepresent the largest two groups. B, Chromatin structure and dynamics; C, Energy production and conversion; D, Cell cycle control, cell division, chromosome partitioning; E, Amino acid transport and metabolism; F, Nucleotide transport and metabolism; G, Carbohydrate transport and metabolism; H, Coenzyme transportand metabolism; I, Lipid transport and metabolism; J, Translation, ribosomal structure and biogenesis; O, Posttranslational modification, protein turnover, chaperones; P, Inorganic ion transport and metabolism; Q, Secondary metabolites biosynthesis, transport and catabolism; R, General function prediction only; T, Signal transduction mechanisms; Z, Cytoskeleton.

genomics data available for this species has hampered characterization of the molecular mechanisms underlying resistance. A deep understanding of the molecular mechanisms regulating insecticide resistance and development may aid in the control of this mosquito by facilitating the development of more sustainable and environmentally friendly approaches. In this study, the proteomics of selected strains with resistance to three distinct chemical families of insecticides (the Cx_cym, Cx_pro, Cx_ddv groups) and the susceptible strain *Cx. pipiens pallens* were compared.

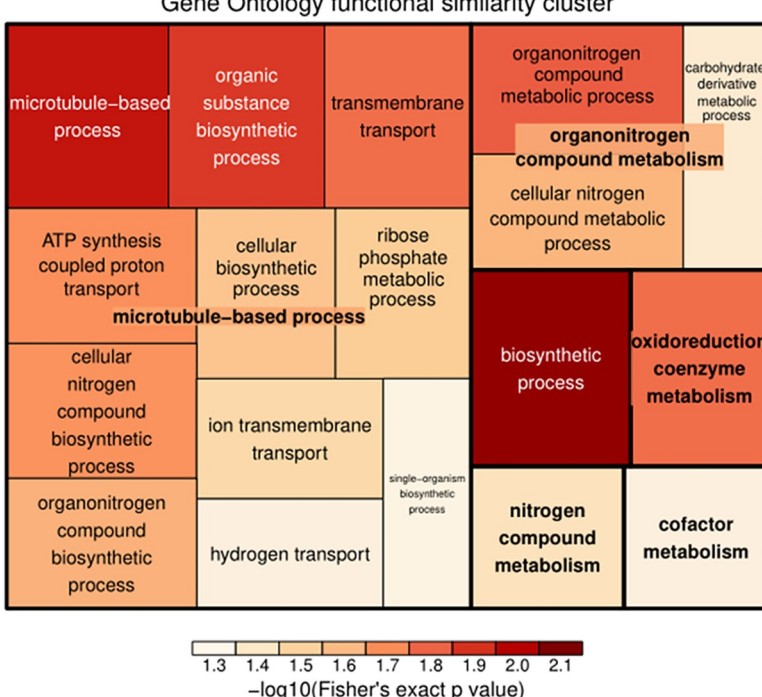

**Fig 5. A treemap overview of significant GO biological processes in the midgut proteome of *Cx. pipiens pallen* exposed with bti.** The diagram shows that exposed with bti condition has a significant effect on proteins biosynthetic process, microtubule-based process, organic substance biosynthetic process, organonitrogen compound metabolic process, oxidoreduction coenzyme metabolic process. The relative sizes of the treemap boxes are based on the |log10 (p-value)| of the respective GO term, related terms are visualized with the same color, the color represents the significant *p* value of this kind of term after -log10, the deeper the color was, the greater the *p* value was, that is, the more significant enrichment the GO terms were.

The insect cuticle, also known as the exoskeleton, is the outermost part of the insect body. It serves a variety of functions, such as sensory perception of the environment, means of locomotion[72], maintenance of the physical structure of the organism, i.e., the foregut, hindgut, tracheal system and apodemes[73], and protection from desiccation; moreover, the cuticle also serves as the first and major barrier against external adverse compound penetration[74]. Thus, the cuticle is a major route of insecticide penetration in insects. Insect cuticles are composed of cuticular protein, chitin and lipids; in fact, the properties of the insect cuticle, including permeability to pyrethroids, are influenced not only by the chitin sclerotization, construction, and hydration but also by regular combinations of different cuticular proteins and their arrangements[74,75]. Recent protein-level and gene-level analyses have demonstrated a surprising diversity of cuticular proteins within and among species[76].

In previous studies, some cuticle genes have been identified as overexpressed in pyrethroid resistant strains of several mosquito species, including *An. stephensi*[77], *An. funestus*[16] and *Cx. pipiens pallens*[20], and the bed bug, *Cimex lectularius*[75,78,79]. The abundance of the protein in limb cuticle correlates nicely with the >2-fold increased abundance of the transcript in pyrethroid resistant *An. gambiae*[32]. The uptake of organophosphate can be reduced by a thickening or a change in the chemical composition of the cuticle in organophosphate-resistant strains of *Culex quinquefasciatus*[80] and *Culex tarsalis*[81]. Pyrethroid-resistant *An. funestus* and *An. gambiae* do indeed have thicker cuticles than sensitive strains[73,82]. Decreased penetration of labeled was observed in the (1R)-trans-permethrin-selected strain of the German cockroach, *Blattella germanica* and in fenvalerate-resistant housefly strains[83,84]. Low levels of

cross-resistance to many insecticides observed in the *An. stephensi* strain might be partially explained by reduced penetration[85]. The epicuticle hydrocarbon lipids were enriched in pyrethroid resistant *An. gambiae*[86] and *An. arabiensis*[87] female mosquitoes.

Our experiments indicate that the GO terms structural constituent of cuticle, structural molecule activity, and chitin binding protein were affected by insecticide selection in *Cx. pipiens pallens*, as a relatively large number of cuticle-associated proteins with a diversity of functions showed highly differential abundance among the cypermethrin-resistant strain, propoxur resistant-strain, DDV-resistant strain and susceptible strain (Figs 2 and 3). In our study, chitin-binding type R&R consensus, insect cuticle protein, actin, conserved site, actin/actin-like conserved site, and actin family were the enriched domains among downregulated proteins in Cx_cym vs. Cx_s. In Cx_pro vs. Cx_s, chitin binding domain proteins were upregulated, whereas insect cuticle protein, chitin-binding type R&R consensus, actin, conserved site, actin/actin-like conserved site, and actin family proteins were downregulated. In Cx_ddv vs. Cx_s, the enriched upregulated proteins were ferredoxin-like fold domain and chitin-binding type R&R consensus, whereas downregulated proteins were insect cuticle protein, chitin-binding type R&R consensus, and adult cuticle protein 1(S3 Fig).

Changes in cuticular protein affect cuticle composition and performance, which is the basis for cuticle-based insecticide resistance. Therefore, other alterations of cuticle structure and composition, such as cuticular remodeling, may also be involved in cuticular resistance. Taken together, our observations confirm that cuticle proteins show altered expression and cellular cytoskeletal rearrangement, suggesting fortification of structural components in response to insecticide selection and leading to increased insecticide resistance in *Cx. pipiens pallens*, likely by producing cuticular-thickening compounds before entering the hemolymph. Thus, cytoskeletion-related proteins are likely major contributors to the three different family insecticides resistance. Although how cuticle proteins are involved in the process of cuticle alterations to slow the penetration of insecticides is puzzling, this result provides useful information for us to conduct further studies on cuticular proteins and the cuticle.

Metabolic resistance occurs via subtle alterations either at the protein level or in the activity of detoxification enzymes that resist insecticides[88,89], such as P450s and GSTs. P450s possess catalytic roles in the metabolism of many essential endogenous molecules and allow insects to metabolize insecticides at a higher rate[26,90]. Up- or downregulation of P450 genes may be responsible for the detoxification of insecticides and the homeostatic response of mosquitoes to changes in the cellular environment [91]. *Cx. pipiens pallens*, similar to most insect species, metabolizes xenobiotics such as insecticides using a suite of detoxification enzymes. We identified the expansion of gene families coding for cytochrome P450s, indicating their predominant roles in the enzymatic detoxification metabolism of chemical insecticides in *Cx. pipiens pallens*. Several detoxification proteins, including cytochrome P450 (B0XDA9), cytochrome P450 3A19 (B0X758), cytochrome P450 4g15 (B0WQV0, B0XHB7), cytochrome P450 9b1 (B0W6Y4), cytochrome P450 monooxygenase CYP9M10 (E5E7H3) and cytochrome P450 12b1 (B0WTS3), have been identified as differentially expressed in insecticide resistant populations in *Cx. pipiens pallens*. In this study, P450 proteins whose expression levels changed significantly belonged primarily to these different families. In addition, data from many other studies have highlighted a key role for P450s in insecticide detoxification in *An. funestus*[16] and *Culex quinquefasciatus*[92,93]. Elevated levels of P450 activity are frequently observed in pyrethroid-resistant malaria vectors in Africa[94–102], in line with data from other insect pests such as the whitefly[103] and cockroach[104]. Overexpression of cytochrome P450s contribute to resistance to neonicotinoid insecticides in the hemipterans *Bemisiatabaci* and *Myzuspersicae*[105,106]. Furthermore, it has been suggested that elevated expression of these P450s also confer cross-resistance to carbamates[95].

Beyond insecticide metabolism and resistance, P450 enzymes play essential roles in hormone metabolism and cuticular hydrocarbon synthesis[107]. Recently, the P450 enzyme responsible for the oxidative decarbonylation of long-chain fatty aldehydes[108] was shown to be a member of the CYP4G subfamily[109], which is restricted to insects, with clearly identifiable sequence peculiarities[109]. Of note, in this present study, cytochrome P450 4g15 (B0WQV0, B0XHB7) belongs to the CYP4G family.

In this study, proteomics sequencing from larvae exposed to insecticide selection (cypermethrin, propoxur and DDV) has confirmed the cross-induction of detoxification enzymes by insecticides; that is, not with particular P450s induced by xenobiotics. Otherwise, detoxification enzymes glutathione S-transferase 1 (B0WQW9, B0XAJ0), glutathione S-transferase 1–1 (B0XGJ8, B0XGK1), glutathione S-transferase (B0VZ90, B0W6D0), glutathione-s-transferase theta (B0XGJ5, B0XGK0, B0XGK3), glutathione synthetase (B0WSC1), glutathione transferase class-sigma-like protein (D3KST8), and glutathione transferase (C4B4V7, C4B4V8) were also identified as differentially expressed in *Cx. pipiens pallens*. GSTs can metabolize insecticides by facilitating their reductive dehydrochlorination or by conjugation reactions with reduced glutathione, to produce water-soluble metabolites that are more readily excreted[110]. Elevated production of GSTs has been documented as a mechanism of resistance in *Spodopteralittoralis*, *Triboliumcastaneum*[111], *Aedes Aegypti*[112], *Musca domestica*[113], *Tenebrio molitor*[111]. Furthermore, the differential expression of glutathione S-transferase (B0W6D0), microsomal glutathione s-transferase (B0X075) and glutathione-s-transferase theta (B0XGK3) enzyme in metabolism of xenobiotics by cytochrome P450 and drug metabolism—cytochrome P450 pathways suggest a critical regulatory role associated with P450s, However, the contribution these GSTs make to insecticide resistance and their biochemical relationships with P450-mediated resistance are still unclear. These results suggested that metabolic insecticide resistance is a complex process involving various enzyme families, including cytochrome P450 and GSTs, with different expression patterns in different resistance strains.

In our study, a number of proteins associated with the ribosome, the primary site of biological protein synthesis that changed in abundance in Cx_cym-vs-Cx_s, Cx_pro-vs-Cx_s and Cx_ddv-vs-Cx_s, including 60S ribosomal protein L18a (B0WTN6), 60S ribosomal protein L7 (B0WRP5), 40S ribosomal protein S15 (B0W2E6), 39S ribosomal protein L43 (B0W3E4), ribosomal protein S6 (B8RIZ4, B0XH45), ribosomal protein L7 (B8RJJ0), all these ribosomal protein with higher increases in insecticides treatment than in the susceptible strain. In addition to the up-regulated ribosomal proteins, we observed decreases in abundance in 40S ribosomal protein S4 (Q152V2), 60S ribosomal protein L10 (B0XGR8), Ribosomal protein L28 (B0WN15), Ribosomal protein L37 (B0WQU6) in Cx_cym-vs-Cx_s, and ribosomal protein S6 (B8RIZ4,) ribosomal protein L7 (B8RJJ0) in Cx_pro-vs-Cx_s. Many ribosome proteins have been linked with cell structure, protein translation and protein biosynthesis[114,115]. Certain ribosomal proteins have functions in regulating cell growth and death in addition to their roles in translation[116]. Our findings confirm the protein level changes of these proteins. The differential expressions of these proteins suggest changes in protein biosynthesis and the process of translating mRNA into protein during mosquito adaptations selection under insecticides. The clear changed expression of ribosomal proteins in our study suggests a possible contribution of these proteins in regulating the insecticides resistance of *Cx. pipiens pallens*.

The tissue-specific expression of a protein is normally related to its function in that tissue [117]. The functional significance of the digestive tract linings in regards to *Cx. pipiens pallens* is especially important. All these tissues contain epidermis and cuticle, as the digestive tract lining constitutes the major part of the body. When *Cx. pipiens pallens* larvae come in contact with or consume insecticides, they may develop resistance by modification of the insect cuticle or digestive tract linings that prevent or reduce the rate of penetration. In this study, cuticular

and cytoskeleton- related proteins are enriched among the proteins with differential expression in the digestive tract linings compared to the Cx_bti strain of *Cx. pipiens pallens*(Figs 4, 5 and S7 and S5 Table). Higher levels of cytochrome P450 3A19 (B0X758) proteins were detected in the midgut in Cx_bti vs. Cx_nbti, suggesting a potential role of P450 in metabolic resistance. Once insecticide enters the *Cx. pipiens pallens* larvae, enhanced metabolic detoxification could decrease the concentration of insecticides before they reach the target site, the midgut-associated *Cx. pipiens pallens* P450s consistent with their role in xenobiotic metabolism.

Our results suggested that a large set of proteins involved in various cellular and metabolic processes were activated and directly or indirectly associated with the insecticide resistance, possibly by influencing the physiology of the cells. In summary, these results significantly increase the molecular resources available for the study of mosquitoes, and we have identified several candidate proteins potentially involved in different phases of insecticide metabolism as well as those participating in other modes of insecticide resistance in *Cx. pipiens pallens*. Significantly differentially expressed proteins across biologically variant samples were revealed by aniTRAQ proteomics approach. Specifically, the high occurrence of differential expression of cuticular and cytoskeleton-related proteins, and the expression patterns of P450s in midgut (cuticular) tissue, support previous studies suggesting unique sites for penetration resistance (cuticle) as well as metabolic resistance (P450s) in *Cx. pipiens pallens*.

A common phenomenon in insecticide resistance is that multiple mechanisms operate simultaneously in resistant mosquitoes. Typically, a combination of diverse mechanisms provides significantly higher levels of resistance than one individual mechanism. In addition, this study has provided further support for the role of P450 genes in insecticide resistance while suggesting that other gene families, such as cuticular genes, could also be involved. Future functional studies of cuticular and P450 proteins could lay the foundation for identifying common sites for insecticide resistance in *Cx. pipiens pallens*, which could provide the basis for developing effective management strategies.

## Supporting information

**S1 Fig.** A. Representative KEGG metabolism of xenobiotics by cytochrome P450 pathway (map00980); B. Representative KEGG drug metabolism—cytochrome P450 pathway (map00982); C. Annotated KEGG map for Oxidative phosphorylation, map00190; D. Annotated KEGG map for Ribosome, map03010 The rectangular nodes in the figure represent gene products, the blue border belongs to the background proteins, and the white color indicates proteins not identified in this experiment. The red/green colors in the figure indicate to the differentially expressed proteins detected in this study, with red representing upregulated proteins and green downregulated proteins. Half red and half green indicates both upregulated and downregulated proteins for that gene product (the same meaning as in this manuscript). (TIF)

**S2 Fig. Comparison of protein expression pattern clusters between Cx_cym, Cx_pro, Cx_ddv and Cx_s in *Cx. pipiens pallens* For the expression trend line graph of each subcluster, the x-axis is the comparison sample group, and the y-axis is the relative expression level of the protein in the group of samples.** Each line in the figure represents a protein, and the different color representations show the relationship between relative expression and the mean value. Each graph shows one type of expression pattern, a trend that reflects changes in the expression of this group of proteins. (TIF)

**S3 Fig. Significantly enriched upregulation and downregulation domain in *Cx. pipiens pallens*, respectively.** Clustering of the results of differential protein enrichment in different groups, the color −log10 (Fisher's exact test *p* value) represents the credibility of enrichment.
(TIF)

**S4 Fig. PPI networks analysis of altered proteins in Cx_cym vs. Cx_s in *Cx. pipiens pallens* (the same meaning as in this manuscript) The interaction map of altered proteins under insecticide selection in Cx. pipiens pallens is illustrated as confidence view, each circle in the figure represents a protein, interaction between expressed proteins are indicated by the connecting line, where the thickness of the connecting lines indicates the level of confidence, stronger associations are represented by thicker lines.** Network interaction analysis was performed using the STRING protein interaction prediction online software and was visualized using Cytoscape version 3.4
(TIF)

**S5 Fig. PPI networks analysis of altered proteins inCx_ddv vs. Cx_s in *Cx. pipiens pallens*.**
(TIF)

**S6 Fig. PPI networks analysis of altered proteins in Cx_pro vs. Cx_s in *Cx. pipiens pallens*.**
(TIF)

**S7 Fig.** A, Significantly enriched domain in Cx_bti vs. Cx_nbti, B, Visualization of significantly enriched downregulation domain in Cx_bti vs. Cx_nbti. Proteins in functional categorizations of proteins differentially expressed according to Fisher's extract test. The Number of Diffproteins is the number of differentially expressed proteins enriched in the Domain; the Domain enriched fold is shown on a log2 scale in a color gradient.
(TIF)

**S1 Table. Resistance levels after insecticide selection.**
(XLSX)

**S2 Table. Proteins identified in *Cx. pipiens pallens* after selection with three distinct chemical families of insecticides using iTRAQ.**
(XLSX)

**S3 Table. Upregulated and downregulated proteins among Cx_cym, Cx_ddv, Cx_pro and Cx_s strains.**
(XLSX)

**S4 Table. PRM verification analysis of insecticide resistance related proteins.**
(XLSX)

**S5 Table. Proteins identified in the midgut of *Cx. pipiens pallens* larvae exposed bti by using iTRAQ.**
(XLSX)

**S1 Data. Number of sequences of downloading.**
(XLSX)

## Acknowledgments

We thank Kai Yu of Shanghai MHelixBioTech Co., Ltd. for their assistance with experiment, we also thank Miao Feng, Haifang Wang (Shandong Institute of Parasitic Diseases) for their review and edit the manuscript and assistance with insect rearing.

## Author Contributions

**Conceptualization:** Chongxing Zhang.

**Data curation:** Tao Li, Xiao Song.

**Formal analysis:** Chongxing Zhang, Tao Li, Xiuxia Guo, Xiao Song.

**Funding acquisition:** Chongxing Zhang, Peng Cheng, Maoqing Gong.

**Investigation:** Chongxing Zhang, Qiqi Shi.

**Methodology:** Chongxing Zhang, Qiqi Shi, Peng Cheng.

**Project administration:** Maoqing Gong.

**Resources:** Qiqi Shi, Peng Cheng, Xiuxia Guo.

**Supervision:** Chongxing Zhang, Maoqing Gong.

**Writing – original draft:** Chongxing Zhang.

**Writing – review & editing:** Chongxing Zhang.

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
