## [Decision Letter · Decision Letter 0]

9 Sep 2020

Dear Dr. Zhang,

Thank you very much for submitting your manuscript "Culex pipiens pallens Coquillett has multi-level molecular resistance mechanism adaptations to different insecticides by comparative proteomics analysis" for consideration at PLOS Neglected Tropical Diseases. As with all papers reviewed by the journal, your manuscript was reviewed by members of the editorial board and by several independent reviewers. In light of the reviews (below this email), we would like to invite the resubmission of a significantly-revised version that takes into account the reviewers' comments. 

The reviewers have indicated the the manuscript is of interest, but significant revisions are necessary. In specific, two commented that along with specific changes the manuscript was difficult to read in many section. Please pay specific attention to comments raised by Reviewer 3.

We cannot make any decision about publication until we have seen the revised manuscript and your response to the reviewers' comments. Your revised manuscript is also likely to be sent to reviewers for further evaluation.

Sincerely,

Joshua B. Benoit

Associate Editor

Karin Kirchgatter

Deputy Editor

The reviewers have indicated the the manuscript is of interest, but significant revisions are necessary. In specific, two commented that along with specific changes the manuscript was difficult to read in many sections. Please pay specific attention to comments raised by Reviewer 3.

Reviewer's Responses to Questions

**Key Review Criteria Required for Acceptance?**

**Methods**

-Are the objectives of the study clearly articulated with a clear testable hypothesis stated?

-Is the study design appropriate to address the stated objectives?

-Is the population clearly described and appropriate for the hypothesis being tested?

-Is the sample size sufficient to ensure adequate power to address the hypothesis being tested?

-Were correct statistical analysis used to support conclusions?

-Are there concerns about ethical or regulatory requirements being met?

Reviewer #1: The authors discussed other mechanisms of insecticide resistance that are far from being fully understood, in addition to resistance at the target site. The work is important in that mosquito control based on chemical insecticides is considered an important element in current global strategies to control mosquito-borne diseases. The authors used a comparative proteomics approach. They have achieved excellent results. I recommend acceptance with minor revision.

But there is still room for improvement in the manuscript.

Reviewer #2: The objectives of the study is clearly articulated with a clear testable hypothesis stated.

I also think the study design is appropriate to address the stated objectives.

The population is nearly described and appropriate for the hypothesis being tested and the sample size is sufficient to ensure adequate power to address the hypothesis being tested and the statistical analysis is also sufficient.

There are no ethical or regulatory requirements being met.

Reviewer #3: Methods are solid, but the presentation of the data should be better edited.

**Results**

-Does the analysis presented match the analysis plan?

-Are the results clearly and completely presented?

-Are the figures (Tables, Images) of sufficient quality for clarity?

Reviewer #1: The legend of the figures can be better improved. Authors should make a detailed description of the colours and meanings of clusters.

Reviewer #2: The analysis presented matches the analysis plan and the results are clearly presented. All figures and tables are very good and have sufficient quality.

Reviewer #3: For clarity, I strongly recommend simplify and edit the KEGG presentations. Please remove unnecessary content. The paper will be then better readable and easier to understand. In the KEGGs there are pathways without any mark/connection – why?

**Conclusions**

-Are the conclusions supported by the data presented?

-Are the limitations of analysis clearly described?

-Do the authors discuss how these data can be helpful to advance our understanding of the topic under study?

-Is public health relevance addressed?

Reviewer #1: I propose a general conclusion of the study that could provide the following perspectives.

Reviewer #2: It is certain that he data support the conclusions. 

The authors also discussed how these data can be helpful to advance our understanding of the topic under study such as offering the viewpoint about using RNA interference on the identified metabolic targets to control those pests.

The relevance of Culex pipiens pallens to public health is addressed well.

Reviewer #3: It can be better presented that a set of markers for future studies was obtained. Novel markers that can be further studied should be better(clearly) stressed. Maybe - section containing "future perspectives" can be useful.

**Editorial and Data Presentation Modifications?**

Reviewer #1: I recommend acceptance with minor revision.

Reviewer #2: An interesting work, it was a pleasure reading. The author used proteome technology to study insecticides-associated resistance mechanism of Culex pipiens pallens and obtained some valuable data, which is helpful to clarify the specific ways and mechanisms of insecticide resistance in mosquitoes. I consider this study has important reference significance for the management of resistance to mosquitoes by using insecticides, and this paper is a valuable contribution to its field.

Below is a list of some comments and changes that might be considered:

1. Title - The title should be made some modification, this manuscript studied the insecticides-associated resistance mechanism by comparative proteomics, is not adaptations to different insecticides, so, I suggest the title revise to ‘Comparative proteomics reveals complex insecticides-associated resistance mechanism of Culex pipiens pallens Coquillett’ is much better.

2. Abstract - The abstract is concise and well documented. I don't have any comments about the same.

3. For KEGG pathways annotation, oxidative phosphophrylation (map00190), ribosome (03010), do you acquire KEGG Copyright Permission? If not, please provide.

4. One of key concerns for the manuscript is that there are some grammatical errors and English improvement need to be carried by professional editor. Some errors but not all that I found are listed below, and I guess there are more mistakes in this manuscript. 

Line 74 ‘resulted in’ can change to ‘result in’

Line 250 ‘increased’ can change to ‘increase’

5. Results - Line 257, A total of 2,502 protein were identified in two replicates of the LC-MS/MS experiments with a high confidence of peptide selection (FDR=0.01). Among them, 1513 protein obtained quantitative message, why were some proteins not quantified?

6. Material - Line 197: What UniProt dataset is used for the annotation? I do not believe a Culex pipiens pallens proteome annotation set exists at this point. Are you using the Culex quinquefasciatus annotation set? The Drosophila melanogaster set? Or something else? 

7. Line 214-217: Same question as above, Culex pipiens pallens is not an available species for KEGG mapping, what dataset is being used?

8. Line 281: What program/algorithm was used to generate the hierarchical clustering? What settings were used?

9. Line 328-332: Again, Culex pipiens pallens is not included in the STRING database, what protein network is used to predict the identified protein interactions? What settings were used for this analysis? 

10. Figure 3: What do the blue highlighted proteins represent? The caption states that “differentially expressed proteins were mapped as red and green”. Is red more abundant and green less abundant? What does it mean when a protein is mapped in BOTH red and green as in figure 3? 

11. Figure 4: What is the y-axis scale in these plots – Log2(fold change), raw fold change?

12. Figure S1: Annotated KEGG map for oxidative phosphorylation, map00190, the caption states “The enzymes marked by the red frame is related to up-regulated gene, and the enzyme labeled by the green frame is related to down-regulated gene.” This raises several questions. First, is this showing gene expression data rather than protein abundance data? If so, where is this data coming from and how does it relate to the proteomics experiment described in the manuscript? Many enzymes are marked in blue, what does this color represent? 

13. Figure S5: What does a figure contribute? It appears to show the entire COG Function Classification map with none of the measured protein data included?

I would recommend that this manuscript could be accepted for publication after minor revisions.

Reviewer #3: (No Response)

**Summary and General Comments**

Reviewer #1: (No Response)

Reviewer #2: (No Response)

Reviewer #3: In this study, Zhang et al. claim the multi-level molecular resistance mechanism adaptations to different insecticides by comparative proteomics analysis in Culex pipiens. Such studies are of importance and interest for number of scientists. I suggest improvements for better readability of the study and possibly stress some more markers. I believe that this interesting study can be published.

For clarity, I strongly recommend simplify and edit the KEGG presentations. Please remove unnecessary content. The paper will be then better readable and easier to understand. In the KEGGs there are pathways without any mark/connection – why?

It is recommended to mention some more relevant studies that investigated the insecticide resistance using proteomics. Authors reference “iTRAQ” study such as doi: 10.1186/s13071-015-0709-5 , however, there are different technologies used, for the proteomic profiling and searching the markers. It would be interesting if the authors could stress more studies that investigated the resistance mechanisms in various insects using the high-throughput proteomics approach – in addition, the introduction could not be limited only to the mosquito. What is the difference (compared to proteomics) if the study is performed on transcript level? or if the resistance is studied based on changes in the DNA sequence (aka the kdr resistance e.g. doi: 10.1371/journal.pone.0215541 https://doi.org/10.1155/2018/3098575)? Authors stress in the conclusion targeting the RNAi. They also stress cuticular, cytochrome P450s and glutathione S-transferase proteins. Indeed, the resistance (especially multiresistance adaptation) can also have the epigenetic basis, which is often overlooked. These mechanisms can be responsible for the multiple resistance mechanisms rather than kdr or the change in CYPs (that respond usually to the specific chemicals-pesticides). See for example DOI: 10.1016/j.jprot.2016.09.016 . If you are able provide discussion on such markers (epigenetic), it can be important for future studies. Resistance may not be “only” kdr, detoxification enzymes (CYPs, GSTs, and other detoxification enzymes such as esterases/lipases, UGTs). As I can see from your results, the ribosomal proteins were importantly changed this appears also of importance, because of translation modulation can be --- linked beyond the ribosome. Further, there is also interesting the vitellinogen change (It may be of importance that there was observed change in vitellogenin isoforms due to deltamethrin treatment 10.1016/j.jprot.2016.09.016 )

Further notes:

- Full names “Bacillus thuringiensis var. Israelensis“are provided at lines 139 and 142 – please revise appropriate use of abbreviations throughout the text

- Line 156: here and elsewhere, please provide x g instead of rpm (rpm does not provide exact info about the centrifugal force because depending on the rotor size)

- Could you please provide the raw data of LC-MS/MS analyses performed?

- Could you please provide number of sequences and the date of downloading? The database can be provided along with the raw LC-MS/MS data.

- Lines 209, 215 – again, you provide full names together with abbreviation “Kyoto Encyclopedia of Genes and Genomes (KEGG)“

- Should “z scores“ should be „Z-score“ ?

- Line 236: proteine-protein - protein-protein

- „involved in cuticular resistence“ – what does it mean?

You mention the “cuticle” / ”cellular cytoskeletal rearrangement“ …. These changes are really hard to explain… However, a proteomic study showed in heads of non-target Bombus terrestris important change in cuticle proteins (in particular cuticle protein 18.7-like ; cuticle protein 21-like in Table 1 of https://doi.org/10.1016/j.jprot.2018.12.022) or the actin binding troponin C and filamin-A due to sublethal insecticide treatment. In that study it was suggested „These changes may reflect partial compensation for the disrupted synthetic pathways, such as the generation of membrane proteins.“ It is possible, that this suggestion may be useful to your story – especially in the discussion (Lines 407-417).

Maybe ??? Is it possible that the resistant insect to pesticides can be differently adapted to the stress linked to changes in the structural proteins?

Indeed, the response of targets as well as non-targets to pesticides appears complex. The tolerance/resistance mechanisms developed can be surprising - as indicated by various high-throughput studies. Unfortunately, some (such) important changes in markers are underestimated in most studies that focus primarily to the detoxification enzymes.

Be aware - the proteins presented as “cuticular/cuticle” the function and localization of the proteins should be confirmed, but most of the proteins are assigned based on similarity. This is important to consider also for number of other proteins. Overall, the changes of the high-throughput analysis give sense together – because some markers change because another markers also change….

PLOS authors have the option to publish the peer review history of their article (what does this mean?). If published, this will include your full peer review and any attached files.

Reviewer #1: No

Reviewer #2: No

Reviewer #3: No
---

## [Decision Letter · Decision Letter 1]

30 Nov 2020

Dear Dr. Zhang,

Thank you very much for submitting your manuscript "Comparative proteomics reveals complex insecticides-associated mechanism of Culex pipiens pallens Coquillett" for consideration at PLOS Neglected Tropical Diseases. As with all papers reviewed by the journal, your manuscript was reviewed by members of the editorial board and by several independent reviewers. The reviewers appreciated the attention to an important topic. Based on the reviews, we are likely to accept this manuscript for publication, providing that you modify the manuscript according to the review recommendations. 

The reviewers have indicated that the manuscript is improved, but still have a few issues. 

1. Along with the specific comments from the reviewers, please ensure that the Dryad link is active. 

2. One of the reviewers commented that the replicate description is not adequate. Please directly indicate that biological replicates used for measuring protein levels. This needs clarification as three lines are mentioned for each selection, so if all lines were examined then at 24 proteomic samples are required to analyze the samples as described (only 8 are shown in table 1). If only a single line is examined twice, this needs to be mentioned as the results are less strong and this requires a correlation analysis between the replicates.

3. The figures need improvement as example, the labeling in Figure 5 and 6 overlap. In addition, it is very likely that figures can be combined. 

Note: Please indicate in the response letter how these points along with those from the reviewers are addressed rather than only indicate that they have been reviewed.

Sincerely,

Joshua B. Benoit

Associate Editor

Karin Kirchgatter

Deputy Editor

The reviewers have indicated that the manuscript is improved, but still have a few issues. 

1. Along with the specific comments from the reviewers, please ensure that the Dryad link is active. 

2. One of the reviewers commented that the replicate description is not adequate. Please directly indicate that biological replicates used for measuring protein levels. This needs clarification as three lines are mentioned for each selection, so if all lines were examined then at 24 proteomic samples are required to analyze the samples as described (only 8 are shown in table 1). If only a single line is examined twice, this needs to be mentioned as the results are less strong and this requires a correlation analysis between the replicates.

3. The figures need improvement as example, the labeling in Figure 5 and 6 overlap. In addition, it is very likely that figures can be combined. 

Note: Please indicate in the response letter how these points along with those from the reviewers are addressed rather than only indicate that they have been reviewed.

Reviewer's Responses to Questions

**Key Review Criteria Required for Acceptance?**

**Methods**

-Are the objectives of the study clearly articulated with a clear testable hypothesis stated?

-Is the study design appropriate to address the stated objectives?

-Is the population clearly described and appropriate for the hypothesis being tested?

-Is the sample size sufficient to ensure adequate power to address the hypothesis being tested?

-Were correct statistical analysis used to support conclusions?

-Are there concerns about ethical or regulatory requirements being met?

Reviewer #1: yes

Reviewer #2: The objectives of the study are clearly articulated with a clear testable hypothesis stated and the study design is appropriate to address the stated objectives doubtlessly； the population clearly described and appropriate for the hypothesis being tested；All the data were statistically analyzed correctly and there are no ethical concerns to consider.

Reviewer #3: (No Response)

**Results**

-Does the analysis presented match the analysis plan?

-Are the results clearly and completely presented?

-Are the figures (Tables, Images) of sufficient quality for clarity?

Reviewer #1: yes

Reviewer #2: The results are clearly and completely presented in the MS and be analyzed correctly. The figures (Tables, Images) are sufficient quality to meet the requirements of publication.

Reviewer #3: (No Response)

**Conclusions**

-Are the conclusions supported by the data presented?

-Are the limitations of analysis clearly described?

-Do the authors discuss how these data can be helpful to advance our understanding of the topic under study?

-Is public health relevance addressed?

Reviewer #1: yes

Reviewer #2: The conclusions are well supported by the data presented and the authors also discuss how these data can be helpful to advance our understanding of the topic under study.

Reviewer #3: (No Response)

**Editorial and Data Presentation Modifications?**

Reviewer #1: accept

Reviewer #2: The author revised the original manuscript with great effort.All concerns have been well addressed and I recommend acccepting this article for publication.However, the author needs to check the text and change the word "drug" with "insecticide",For example, a similar error occurs in Figure 3.b.

Reviewer #3: (No Response)

**Summary and General Comments**

Reviewer #1: no comment.

Reviewer #2: Proteomics is undoubtedly a powerful tool for studying the mechanisms of insect resistance.This study has found a number of interesting proteins, some of which are clearly reported to be involved in resistance and some of which are not, and it is hoped that subsequent studies will clarify these important protein functions and the association between resistance.It maybe provide better targets for insecticide resistance detection and management.

Reviewer #3: The manuscript has been improved and I am respecting the way of presentation. Authors must be aware that only minor part of the proteins were COG classified. Thus, by that connection it is of high importance to stress and discuss the particular markers. It is a pity that authors did not improve the discussion despite recommendations in previous version. It is a pity that instead of discussion of particular markers with different studies the section about cuticular proteins (now lines 484-496 and 497-511) appears only general. It is recommended to provide discussion on particular markers and this can be done as minor revision.

In addition, would like to see clustering on columns – Fig 3. How do the treatments cluster?

Lines231+ Please provide number of sequences used for search along with the date and source of download.

PLOS authors have the option to publish the peer review history of their article (what does this mean?). If published, this will include your full peer review and any attached files.

Reviewer #1: Yes: Abdou Azaque ZOURE

Reviewer #2: No

Reviewer #3: No
---

## [Editor Report · Decision Letter 2]

9 Dec 2020

Dear Dr. Zhang,

Thank you very much for submitting your manuscript "Comparative proteomics reveals complex insecticides-associated mechanism of Culex pipiens pallens Coquillett" for consideration at PLOS Neglected Tropical Diseases. As with all papers reviewed by the journal, your manuscript was reviewed by members of the editorial board and by several independent reviewers. The reviewers appreciated the attention to an important topic. Based on the reviews, we are likely to accept this manuscript for publication, providing that you modify the manuscript according to the review recommendations. 

The manuscript is improved, but there are still quality issues with figures and in the manuscript. 

1. Labels need to be added to the scale for each figure. 

2. Parts of the figures are cut off. 

3. Labels are overlapping in Figure 6. 

4. Format references. 

5. Please check the manuscript for grammar and format issues.

Sincerely,

Joshua B. Benoit

Associate Editor

Karin Kirchgatter

Deputy Editor

The manuscript is improved, but there are still quality issues with figures and in the manuscript. 

1. Labels need to be added to the scale for each figure. 

2. Parts of the figures are cut off. 

3. Labels are overlapping in Figure 6. 

4. Format references. 

5. Please check the manuscript for grammar and format issues.
---

## [Editor Report · Decision Letter 3]

13 Jan 2021

Dear Dr. Zhang,

Thank you very much for submitting your manuscript "Comparative proteomics reveals complex insecticides-associated mechanism of Culex pipiens pallens Coquillett" for consideration at PLOS Neglected Tropical Diseases. As with all papers reviewed by the journal, your manuscript was reviewed by members of the editorial board and by several independent reviewers. The reviewers appreciated the attention to an important topic. Based on the reviews, we are likely to accept this manuscript for publication, providing that you modify the manuscript according to the review recommendations. 

There are still issues that persist in the manuscript. 

1. Please correct the references by hand. There are many errors, specifically species names should be in italics and other aspects of the references aren't uniform. 

2. The scale labels are still missing. There are not units. Are these fold change, z scores, etc?

3. Figure 6: Revigo will output an editable pdf. Labels should not overlap. Also, what setting were used for this program. 

4. All abbreviations need to be defined in the figure legends. 

5. Some of the figure legends lack details. For example, the title does not necessarily reflect what is described in the legend. Please review all figure titles and legends. 

6. What do the cluster represent in Fig. 4? This is not integrated well into the results. 

Following these revisions, please provide a direct response to these specific comments (if possible) with line numbers.

Sincerely,

Joshua B. Benoit

Associate Editor

Karin Kirchgatter

Deputy Editor

There are still issues that persist in the manuscript. 

1. Please correct the references by hand. There are many errors, specifically species names should be in italics and other aspects of the references aren't uniform. 

2. The scale labels are still missing. There are not units. Are these fold change, z scores, etc?

3. Figure 6: Revigo will output an editable pdf. Labels should not overlap. Also, what setting were used for this program. 

4. All abbreviations need to be defined in the figure legends. 

5. Some of the figure legends lack details. For example, the title does not necessarily reflect what is described in the legend. Please review all figure titles and legends. 

6. What do the cluster represent in Fig. 4? This is not integrated well into the results. 

Following these revisions, please provide a direct response to these specific comments (if possible) with line numbers.
---

## [Editor Report · Decision Letter 4]

1 Feb 2021

Dear Dr. Zhang,

Thank you very much for submitting your manuscript "Comparative proteomics reveals complex insecticides-associated mechanism of Culex pipiens pallens Coquillett" for consideration at PLOS Neglected Tropical Diseases. As with all papers reviewed by the journal, your manuscript was reviewed by members of the editorial board and by several independent reviewers. The reviewers appreciated the attention to an important topic. Based on the reviews, we are likely to accept this manuscript for publication, providing that you modify the manuscript according to the review recommendations. 

Response to previous comments are not sufficient.

1. There is still no indication that the scales for each figure are fold change, z score, etc. This still needs correction in all figures. As an example, there is no label on scale bar for Figure 1 and no indication what the scale represents in the legend. This needs correction as the reader would have no idea what this figure represents based on Figure 1 and the associated legend. Similarly, the scale labels should be moved next to the scale bar in Figure 2 and Figure 3. 

2. The figure legends still lack specific details to be read independently, such as abbreviations should be discussed, what is actually significantly upregulated, bold vs. normal font differences, and any other details. Also, please delete "Visualization of" as this is redundant. 

3. Fisher is not spelled correctly in Figure 5. 

4. For clarity purposes, please move the Figure legends to the end of the document.

Sincerely,

Joshua B. Benoit

Associate Editor

Karin Kirchgatter

Deputy Editor

Response to previous comments are not sufficient.

1. There is still no indication that the scales for each figure are fold change, z score, etc. This still needs correction in all figures. As an example, there is no label on scale bar for Figure 1 and no indication what the scale represents in the legend. This needs correction as the reader would have no idea what this figure represents based on Figure 1 and the associated legend. Similarly, the scale labels should be moved next to the scale bar in Figure 2 and Figure 3. 

2. The figure legends still lack specific details to be read independently, such as abbreviations should be discussed, what is actually significantly upregulated, bold vs. normal font differences, and any other details. Also, please delete "Visualization of" as this is redundant. 

3. Fisher is not spelled correctly in Figure 5. 

4. For clarity purposes, please move the Figure legends to the end of the document.
---

## [Editor Report · Decision Letter 5]

12 Feb 2021

Dear Dr. Zhang,

We are pleased to inform you that your manuscript 'Comparative proteomics reveals complex insecticides-associated mechanism of Culex pipiens pallens Coquillett' has been provisionally accepted for publication in PLOS Neglected Tropical Diseases.

Best regards,

Joshua B. Benoit

Associate Editor

Karin Kirchgatter

Deputy Editor

I am happy to recommend for publication, but please check the manuscript carefully before publication:

1. Please check carefully for errors (e.g. there is a misspelling in the Figure 4 title, which is also confusing, Line 39: there is a word missing, and others).

2. I would recommend that the title be changed as "Comparative proteomics reveals complex insecticides-associated mechanism of Culex pipiens pallens Coquillett" is not correct. I would suggest "Comparative proteomics reveals mechanisms that underlie insecticide

resistance in Culex pipiens pallens Coquillett"

---

## [Editor Report · Acceptance letter]

3 Mar 2021

Dear Dr. Zhang,

We are delighted to inform you that your manuscript, "Comparative proteomics reveals mechanisms that underlie insecticide resistance in Culex pipiens pallens Coquillett," has been formally accepted for publication in PLOS Neglected Tropical Diseases.

Best regards,

Shaden Kamhawi

co-Editor-in-Chief

Paul Brindley

co-Editor-in-Chief
